# Diffusion Behavior of Iodine in the Micro/Nano-Porous Graphite for Nuclear Reactor at High Temperature

**Ming-Bo Qi** [1,2,†], **Peng-Fei Lian** [3,4,†], **Peng-Da Li** [2,4], **He-Yao Zhang** [1], **Jin-Xing Cheng** [5], **Qing-Bo Wang** [5], **Zhong-Feng Tang** [2,6], **T. J. Pan** [1,*], **Jin-Liang Song** [2,6,*] and **Zhan-Jun Liu** [3,6,*]

1   School of Materials Science and Engineering, Changzhou University, Changzhou 213164, China; qimingbo@sinap.ac.cn (M.-B.Q.)
2   Shanghai Institute of Applied Physics, Chinese Academy of Sciences, Shanghai 201800, China
3   Key Laboratory of Carbon Materials, Institute of Coal Chemistry, Chinese Academy of Sciences, Taiyuan 030001, China; lianpengfei0302@163.com
4   University of Chinese Academy of Sciences, Beijing 100049, China
5   Beijing High-Tech Institute, Beijing 100094, China
6   Dalian National Laboratory for Clean Energy, Dalian 116023, China
*   Correspondence: tjpan@cczu.edu.cn (T.J.P.); jlsong1982@yeah.net (J.-L.S.); zjliu03@sxicc.ac.cn (Z.L.)
†   These authors contributed equally to this work.

**Abstract:** The diffusion behavior of iodine in micro/nano-porous graphite under high-temperature conditions was studied using analysis methods such as Rutherford backscattering Spectrometry, scanning electron microscopy, X-ray diffraction, and Raman spectroscopy. The results indicate that iodine diffusion leads to the Lattice Contractions in Microcrystals, a decrease in interlayer spacing, and a rise of defect density. And the reversal or repair of microstructure change was observed: the microcrystal size of the graphite increases, the interlayer spacing appears to return to the initial state, and the defect density decreases, upon diffusion of iodine out of iodine-loaded graphite. The comparative study comparing the iodine diffusion performance of nanoporous graphite (G400 and G450) between microporous graphite (G500), showed that nanoporous graphite exhibits a better barrier to the iodine diffusion. The study on the diffusion behavior of iodine in micro/nano-porous graphite holds substantial academic and engineering value for the screening, design, and performance optimization of nuclear graphite.

**Keywords:** micro/nano-porous graphite; iodine; diffusion; Rutherford backscattering spectrometry





## 1. Introduction

Diffusion of radioactive isotopes of iodine, particularly [131]I, in High-Temperature Gas-cooled Reactors (HTGRs) deserves attention both during normal operation and accident scenarios. When acquiring experience on iodine diffusion behavior in such reactors, the diffusion mechanisms between iodine and core materials must be taken into account [1]. And this is crucial for assessing the durability of core materials and optimizing their performance, affected by the fission product interaction with the core materials, under the influence of the interaction between fission products and core materials. Graphite, which serves as both a moderator and structural material, is a key material in HTGR cores [2]. Investigating the diffusion behavior of iodine in graphite holds significant scientific importance for reactor design. Given the current development of graphite products, isotropy and uniform graphite with dense structure, low porosity materials such as G400, G450, and G500 as potential candidates for various HTGR designs. This paper mainly analyzes the diffusion behavior of iodine in micro/nano-porous graphite at high temperatures.

Currently, the diffusion behavior of iodine in graphite has not been well characterized. Between 1976 and 1997, Muller reported on the migration of iodine in A3 matrix graphite during normal operation of HTGRs, describing the mechanisms of iodine diffusion through

pores or graphite particle diffusion and further explaining that the rapid release of iodine from graphite is due to fast transport through the pores [3,4]. However, due to limitations in graphite manufacturing technology at that time, the graphitization degree of A3 matrix graphite was relatively low and did not meet the design requirements of HTGRs. On the other hand, IG-110 graphite with an average pore size of 2 μm has become an ideal candidate material for reactors due to its higher graphitization degree, high temperature stability, strength and stiffness, low thermal expansion coefficient, and corrosion resistance. Therefore, Carter [5,6] tested the diffusion coefficients of $I^-$ and $I_2$ in bulk IG-110 graphite and compacted IG-110 graphite using ICP-MS, and the results showed that there was no significant difference in iodine diffusion coefficients between bulk IG-110 and compacted IG-110 graphite, both at the order of 10–10. They further described the low sensitivity of iodine diffusion to graphite pore size. In addition, Mukhawana [1] reported on the behavior of iodine injected into heat-treated highly oriented pyrolytic graphite (HOPG), where the results indicated an increased iodine loss rate with increasing temperature. Although there have been some reports on the migration and diffusion of iodine in graphite, there is still a lack of a systematic understanding of iodine diffusion behavior in graphite, particularly in micro/nano-porous graphite.

In this work, micro/nano-porous graphite G400, G450, and G500 with average pore sizes of 23 nm, 18 nm, and 553 nm, respectively, were used as the subjects for diffusion experiments. Iodine vapor at 650 °C was employed as the diffusion source. The diffusion behavior of iodine in micro/nano-porous graphite was investigated using Rutherford Backscattering Spectrometry (RBS), Scanning Electron Microscopy (SEM), X-ray Diffraction (XRD), and Raman spectroscopy. The aim was to elucidate the diffusion mechanisms of iodine in micro/nano-porous graphite and to provide important scientific insights for the screening, design, and optimization of nuclear graphite materials.

## 2. Experimental

### 2.1. Sample Preparation

Diffusion behavior of iodine in micro/nano-porous graphite was investigated by iodine vapor diffusion method. A series of micro/nano-porous graphite as research objects include both nanoporous graphite G400 and G450, as well as microporous graphite G500. The micro/nano-porous graphite derived from the mixtures of coal-tar pitch, natural graphite flake (NGF) and tetrahydrofuran was prepared by a liquid-phase mixing processsaccording to our previous work [7]. G400, G450 and G500 were named according to calcination temperature of the calcined mixed materials (calcined mixed material equivalent to coke). The pore size distribution and mercury intrusion curves of the three types of graphite are shown in Figure 1a and 1b, respectively, and their properties are summarized in Table 1. The two nanoporousgraphites (G400 and G450) have similar median pore sizes, open porosities, and densities. The large pores near 10 μm in G400 originate from the fact that the density of the green body is too high to release a large number of volatiles, which leads to some large cracks in G400. And the open porosity of micron porous graphite (G500) is lower than that of nanoporous graphite. Figure 1b shows the intrusive mercury curve of G400, G450, and G500. As is well known, threshold pressure plays an important role in the mercury intrusion process. Before the pressure increases to the threshold pressure, only a small amount of mercury can be injected, and this mercury injection process is the first stage. Once the pressure increases near the threshold pressure and continues to increase, a large amount of mercury will be injected into the pores of the sample, and this mercury injection process is the second stage. After the second stage, as the pressure increases, mercury may continue to be injected into the sample, as the injectable pores may continue to increase due to mechanical rupture of the closed pores. This mercury injection process is the third stage. In addition, it may be difficult to distinguish between the second and third stages when the threshold pressure is high enough. The threshold pressure is closely related to the pore size distribution of the sample. The smaller the pore size of the sample, the higher the threshold pressure, and the more difficult it is to distinguish between the

second and third stages. Based on the above discussion, the main pore sizes of G400 and G450 are less than 30 nm, while the main pore sizes of G500 are greater than 500 nm. The average pore size of G400 and G450 is 23 nm and 18 nm, respectively, while the average pore size of G500 is 553 nm. Properties of micro/nano-porous graphite such as density, porosity or graphitization are shown in Table 1. In this work, G400, G450 and G500 were cut and ground into rectangular sheets with a size of $7.0 \times 7.0 \times 1.0$ mm$^3$. The instruments and operations used for cutting and polishing all samples in this experiment were kept consistent to minimize experimental errors. The processed rectangular graphite sheets were repeatedly rinsed through anhydrous ethanol and deionized water, and then dried for iodine diffusion experiments.

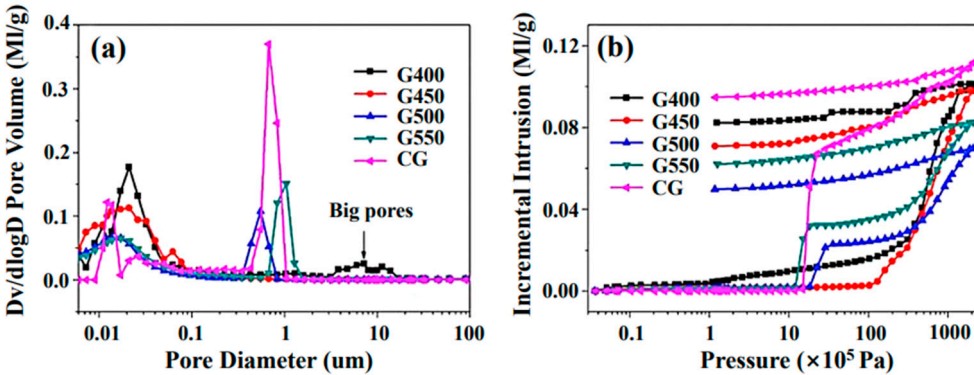

**Figure 1.** (**a**,**b**) show the pore size distribution curves and mercury compression curves for G400, G450 and G500, respectively. G550 and CG are prepared from previous work and are not described in this paper [7].

**Table 1.** Properties of micro/nano-porous graphite.

| Properties | G400 | G450 | G500 |
|---|---|---|---|
| Apparent density (g/cm$^3$) | $1.76 \pm 0.02$ | $1.77 \pm 0.02$ | $1.84 \pm 0.02$ |
| Graphitization degree (%) | $91.5 \pm 0.2$ | $92.0 \pm 0.2$ | $93.8 \pm 0.2$ |
| Average pore size (volume, nm) | 23 | 18 | 553 |
| Open porosity (%) | $17.8 \pm 0.1$ | $17.3 \pm 0.1$ | $12.8 \pm 0.1$ |

*2.2. Diffusion Experiment*

In this work, the iodine (Sinopharm Chemical Reagent Co., Ltd., Shanghai, China) was purchased from Sinopharm Chemical Reagent Co., Ltd. with a purity of ≥99.99% and a total number of magazine cations ≤0.01%.The details of the iodine diffusion experiment are described below. First, Iodine pellets (mass of about 0.40 g) were placed into dehydroxy-lated quartz tubes (the purpose of high temperature annealing of quartz tubes is to remove the hydroxyl groups from the quartz tubes themselves to prevent them from affecting the experiment). And the processed $7.0 \times 7.0 \times 1.0$ mm$^3$ micro/nano-porous graphite samples G400, G450 and G500 were immediately packed into quartz tubes. Then the quartz tubes were placed on a vacuum sealer for sealing and packing, while the air was removed from the quartz tubes. At the same time, the vacuum inside the sealed quartz tube is 0.15 Pa. Next, the above steps were repeated until the G400, G450 and G500 graphite were sealed in three quartz tubes, respectively. It is worth noting that the purpose of maintaining a vacuum inside the quartz tubes is to reduce the interference of airborne components in the work, especially oxygen and water. Finally, the quartz tube loaded with graphite and iodine was placed in GSL-1700X-HV high vacuum tube furnace(Hefei Kejing Material Technology Co., Ltd., Hefei, China), and then the high temperature diffusion experiment at 650 °C was performed. Figure 2a shows the experimental schematic diagram of the diffusion process of G400, G450 and G500 in iodine vapor at 650 °C for 96 h. And Figure 2b shows a schematic diagram of the experimental process of isothermal annealing of G400, G450 and G500

graphite loaded with iodine for 48 h. It is noteworthy that the isothermal annealing process was performed after the micro/nano-porous graphite had diffused in iodine vapor for 96 h and the iodine-loaded micro/nano-porous graphite was re-encapsulated in a quartz tube. The purpose of isothermal annealing is to study the process of iodine desorption and diffusion. Graphite samples were obtained by the diffusion process described above. The naming details of the experimental samples are as follows: graphite that were not subjected to diffusion experiments were named after their original names (G400, G450, and G500), graphite samples diffused under iodine vapor at 650 °C for 96 h were named with the addition of the '-D96' suffix (G400-D96, G450-D96, and G500-D96), and graphite samples that were loaded with iodine after the above steps and isothermally annealed for 48 h were named with the addition of the '-H48' suffix (G400-H48, G450-H48, and G500-H48).

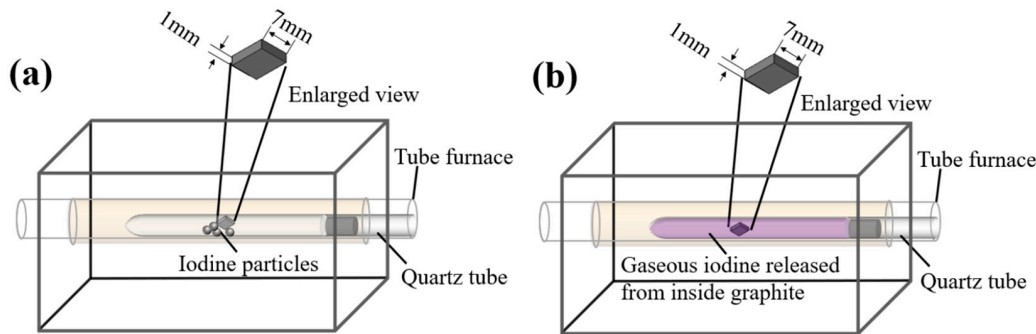

**Figure 2.** Schematic diagram of iodine diffusion experiments: (**a**) diffusion of micro/nano-porous graphite in iodine vapor at 650 °C for 96 h; (**b**) isothermal annealing of iodine-loaded micro/nano-porous graphite for 48 h.

### 2.3. Characterizations

The morphology and the structure changes of the graphite samples were monitored using SEM (CarlZeissAG, Oberkochen, Germany) and their structures before and after iodine diffusion were measured using Bruker D8 Advance XRD (Bruker, Karlsruhe, Germany) with CuKα1 radiation source ($\lambda$ = 1.5406 Å).The radiation source is conditioned by two 2.5° Soller slits and a 0.025 mm Ni mask. Rutherford scattering spectrometry (RBS, Shanghai Institute of Applied Physics, Chinese Academy of Sciences, Shanghai, China) was used to test the depth of diffusion of iodine in micro/nano-porous graphite by a Van de Graaff accelerator and a 2.0 Mev He$^+$ beam. The angle of incidence of helium ions is 165° and the spot of the test element signal is 1 μm. The reflected X-ray intensities were collected by a LynxEye XE counter using continuous theta-2theta scans at a tube power of 40 kV/40 mA, ranging from 10° to 90° (2theta) with steps of 0.02° (2theta) at 0.15 s intervals. Changes in defects caused by iodine diffusion were recorded using a Raman spectrometer (Horiba Jobin Yvon, Paris, France) at an excitation wavelength of 473 nm and an effective penetration depth of about 50 nm.

## 3. Results and Discussion

### 3.1. Microstructure

The porous microstructure of graphite consists of microcracks, open pores and closed pores at a macroscopic scale. Figure 3a–c show the surface structure of micro/nano-porous graphite before iodine diffusion. Nanoporous graphite G400 and G450 have lower roughness than microporous graphite G500, while the surface structure of G450 is denser than that of G400. Figure 3d–f show the SEM images of graphite surfaces of G400, G450, and G500 diffused in iodine vapor at 650 °C for 96 h. Compared with the initial surface morphology, the surface of the diffused graphite becomes rougher. But the surface texture of G450 graphite is not as clear as that of G400 and G500, because G450 graphite is least affected by iodine vapor (G450 has the smallest average pore size). Figure 3g–i show iodine-loaded micro/nano-porous graphite undergoing isothermal annealing for 48 h at

the same temperature and vacuum to observe the diffusion of iodine from the inside of the graphite to the surface. Interestingly, iodine was precipitated near the pores on the surface of graphite, after the annealed graphite was placed in vacuum at room temperature for a period of time, which indicates that pores are an important channel for iodine diffusion.

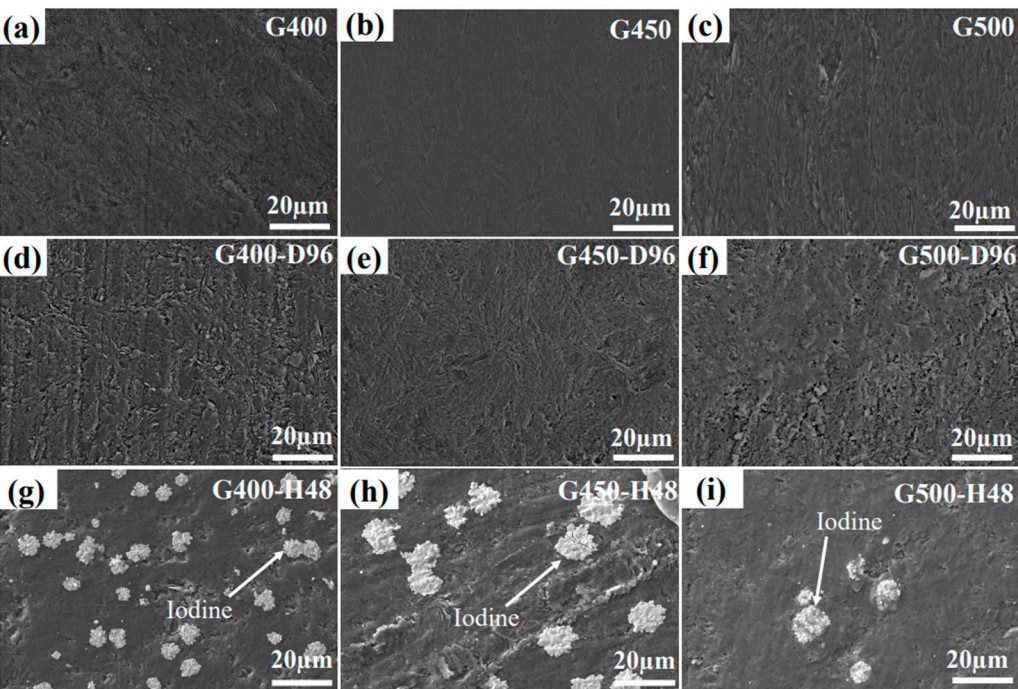

**Figure 3.** Surface SEM micrographs of pristine micro/nano-porous graphite (**a**–**c**), iodine diffusion in the graphite at 650 °C for 96 h (**d**–**f**) and isothermal annealing of the graphite loaded with iodine for 48 h (**g**–**i**).

Random cracks and pores could be identified in the micro/nano-porous graphite of the pristine structure shown in Figure 4. Figure 4a,b show large longitudinal pores inside G400, which are generated by the interaction between volatile gas escape during graphitization and anisotropic contraction of graphite crystals during the cooling phase of graphite manufacturing. Compared with G400 and G500, nanoporous graphite G450 has a smooth surface and small pore size. However, G450's surface occasionally has scaly transverse pores. The pores on the surface of G500 are large and uniform, and these pores are caused by volatiles in the adhesive during the sintering process.

Figure 5 shows the EDS image of G400, G450 and G500 graphite diffused in iodine vapor for 96 h at 650 °C, respectively. The distribution of iodine diffusion is on the graphite surface, and the blues and the reds in Figure 5b,c,e,f,h,i represent carbon and iodine, respectively. The relative content of iodine in G400, G450 and G500 are the same as that of energy dispersive spectrometer (EDS) analysis, as shown in Table 2. Since the diffusion of gaseous substances generally includes two processes of adsorption and diffusion, the smooth graphite surface is not easy to adsorb and diffuse substances. Iodine vapor is not easily attached to the smooth G450 graphite surface, so the iodine content on the surface of G450 is smaller than that of G400 and G500.

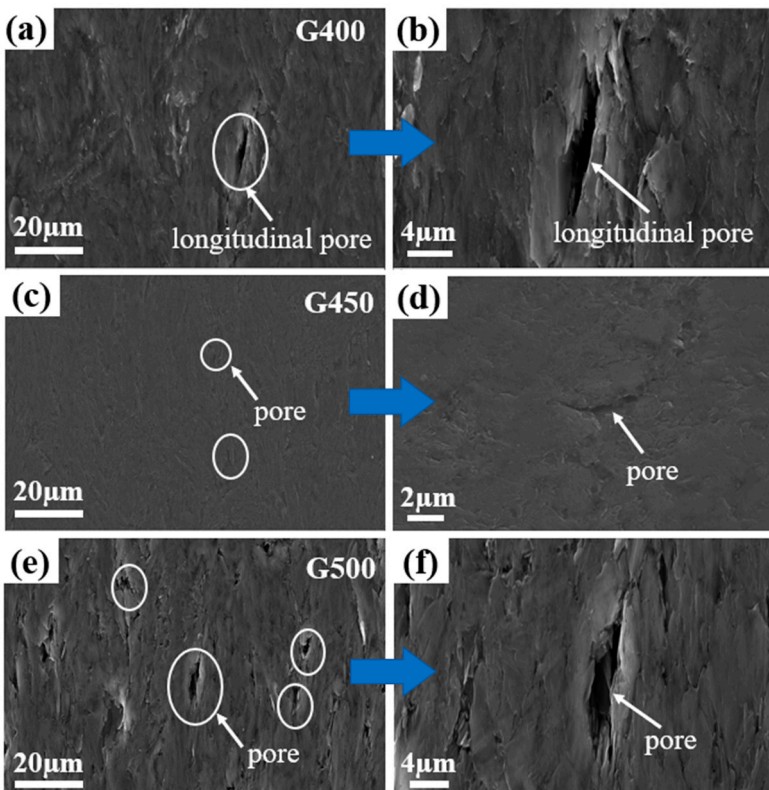

**Figure 4.** (**a**,**c**,**e**) represent SEM images of the microscopic pores on the surface of G400, G450, and G500 graphite, respectively, while (**b**,**d**,**f**) correspond to enlarged images of their pores, respectively. The inside of the circle shows the pores generated by the volatilization of volatile substances during the graphite manufacturing process. The arrow indicates magnifying the pore.

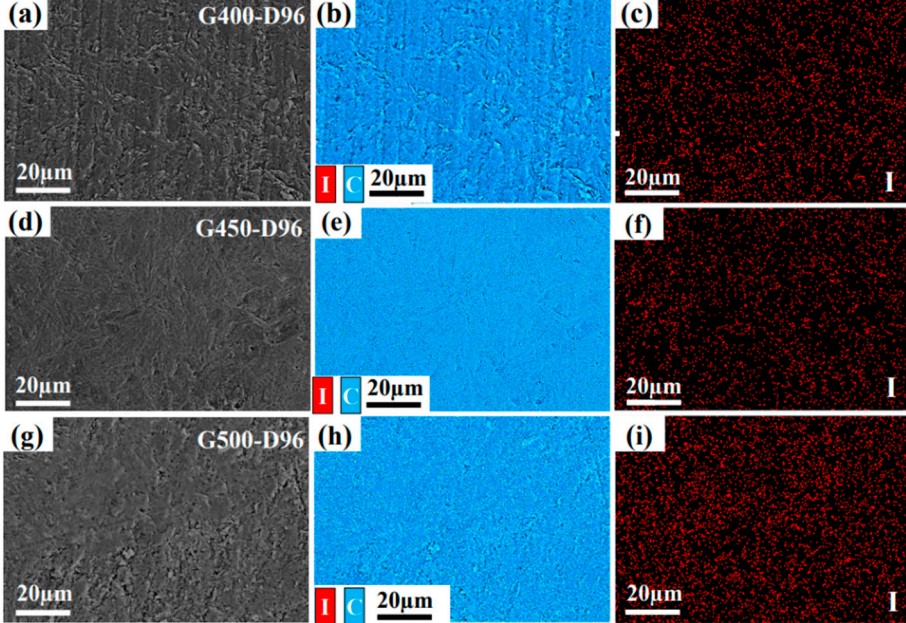

**Figure 5.** (Color online) Iodine distribution on the surface of G400, G450 and G500 graphite is respectively diffused in iodine vapor at 650 °C for 96 h. (**a**,**d**,**g**) are the SEM image, and (**b**,**c**,**e**,**f**,**h**,**i**) are EDS image (the blue and the red color represent the distribution of C and I, respectively).

**Table 2.** The relative content of iodine and carbon on the surface of micro/nano-porous graphite after the diffusion experiment.

| Element | G400-D96 | G450-D96 | G500-D96 |
|---|---|---|---|
| Carbon (%) | 96.70 ± 0.02 | 97.20 ± 0.02 | 96.10 ± 0.02 |
| Iodine (%) | 3.30 ± 0.02 | 2.80 ± 0.02 | 3.90 ± 0.02 |

*3.2. Rutherford Backscattering Spectrometry Analysis*

Rutherford backscattering spectrometry (RBS) is an efficient analytical method for detecting the diffusion depth of elements with higher atomic numbers in the periodic table in a matrix material for elements with lower atomic numbers [8,9]. Figure 6a shows the RBS experimental spectra of micro/nano-porous graphite (G400, G450, and G500) after diffusion in iodine vapor at 650 °C for 96 h. The respective surface channel positions of C and I are indicated by arrows, and the distribution of I presents a distribution in which the concentration gradually decreases from the surface to the interior. Figure 6b shows the depth profiles of micro/nano-porous graphite (G400, G450, and G500) after diffusion in iodine vapor at 650 °C for 96 h. The difference in iodine distribution between nanoporous graphite G400 and G450 and microporous graphite G500 is shown in Figure 6a,b. The RBS experimental spectrum shows that after diffusing in iodine vapor at 650 °C for 96 h, the iodine content of G450 and G500 gradually decreases from the graphite surface to the interior of the matrix, while the iodine content of G400 has little difference between the surface and interior of graphite. The iodine content on the surface of G450 is higher than that of G400 and G500, which seems to contradict the EDS analysis results that the iodine content on the surface of G450 is the least. The implementation is just the opposite, since the diffusion of gaseous species is related to the pore size and cracks of graphite. Nanoporous graphite G450 has a smaller average pore size, less defects such as cracks, and its surface is smooth. Although G400 graphite is also nanoporous graphite (average pore size is 23 nm), its surface has cracks and large longitudinal pores. These larger cracks and longitudinal pores make the connected pore network on the surface of G400 larger (17.8% open porosity), resulting in little difference between its surface and internal iodine. These data results indicate that larger cracks and longitudinal pores propagate more easily. Both G450 and G500 graphite have a common point that the pore distribution is relatively uniform. The difference is that the open porosity of G450 (17.3%) is higher than that of G500 (12.8%),while the average pore diameter of G500 (553 nm) is bigger than that of G450 (18 nm). It can be seen from Figure 6a,b that there is more iodine on the surface of nanoporous graphite G450 with a depth of less than 400 nm than that of G500, because the graphite surface with a larger open porosity is more likely to capture iodine. But at depths greater than 400 nm, the iodine distribution of both nanoporous graphite G400 and G450 is lower than that of microporous graphite G500. The iodine distribution of G450 graphite is the smallest of the three graphite, which is caused by its smaller average pore size and fewer frontal defects. In short, nanoporous graphite G450 has better performance of blocking iodine diffusion in the process of adsorption and diffusion.

Figure 7a shows the experimental RBS spectra of iodine-loaded micro/nano-porous graphite (G400, G450 and G500) annealed for 48 h under vacuum at 650 °C to describe the desorption and diffusion behavior of iodine-loaded graphite. Figure 7b shows the depth distribution of iodine, and the surface of C is marked with arrows. Annealing causes iodine to diffuse deeper into the graphite. Since part of the iodine on the graphite surface is rapidly released during annealing, and the internal iodine continues to diffuse to the surface, a wide iodine peak appears in all three kinds of graphite at 650 nm from the graphite surface. However, the iodine peak of microporous graphite G500 at 650 nm is not pronounced due to the formation of a stable iodine diffusion channel inside the G500 of the larger pore [10]. At depths greater than 1.4 µm, nanoporous graphite G400 and G450 have less iodine than microporous graphite G500, indicating that pore size is a key factor affecting iodine diffusion performance, and nanoporous graphite has better performance

in blocking iodine diffusion than microporous graphite. Combined with the results of SEM surface morphology analysis, the nanoporous graphite G450 has best barrier performance to iodine at high temperature owing to smaller pores, fewer cracks and smoother surface.

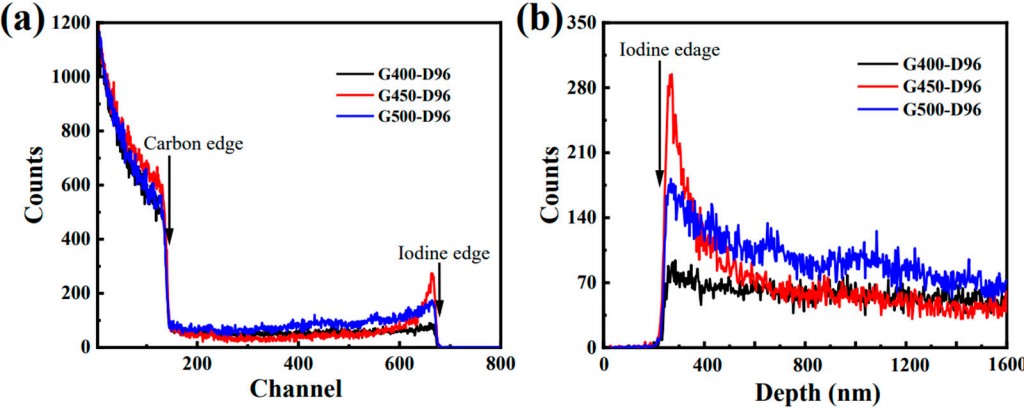

**Figure 6.** (**a**) RBS spectrum of iodine diffused in micro/nano-porous graphite at 650 °C for 96 h. (**b**) RBS depth profile of iodine diffused at 650 °C for 96 h.

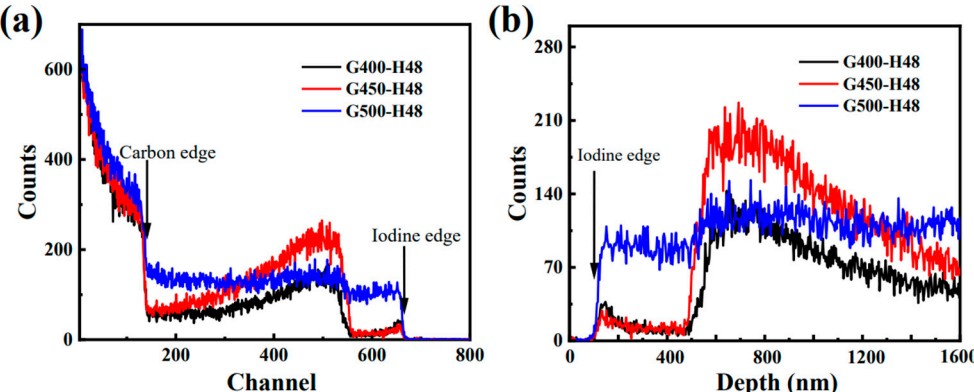

**Figure 7.** (**a**) RBS spectrum of iodine desorbed in micro/nano-porous graphite at 650 °C for 48 h; (**b**) RBS depth profile of iodine desorbed at 650 °C for 48 h.

### 3.3. Crystal Structure

In order to illustrate the effect of iodine diffusion on graphite microstructure, the diffraction angle of 002 plane and Bragg formula were used to analyze the variation of graphite interlayer spacing. According to the Bragg formula $2d\sin\theta = n\lambda$ and $d_c = (d - d_1)/d$ (where $d_c$ is the change in interlayer spacing, d is the interlayer spacing of the original graphite, and $d_1$ is the graphite interlayer spacing after iodine diffusion) to calculate the variation of graphite interlayer spacing before and after diffusion. Figure 8a–c show the X-ray diffraction (XRD) patterns of micro/nano-porous graphite (G400, G450 and G500) of different experimental conditions, including the change trend of micro/nano-porous graphite (002) peak, and the change trend of micro/nano-porous graphite interlayer spacing under different heat treatment time, respectively. Figure 8a shows that the (111) peak near the (002) peak indicated the iodine diffuses into micro/nano-porous graphite at 650 °C, which is similar to the peaks that appear near the (002) peak of graphite in the literature due to the presence of silicon impurities [11–13].

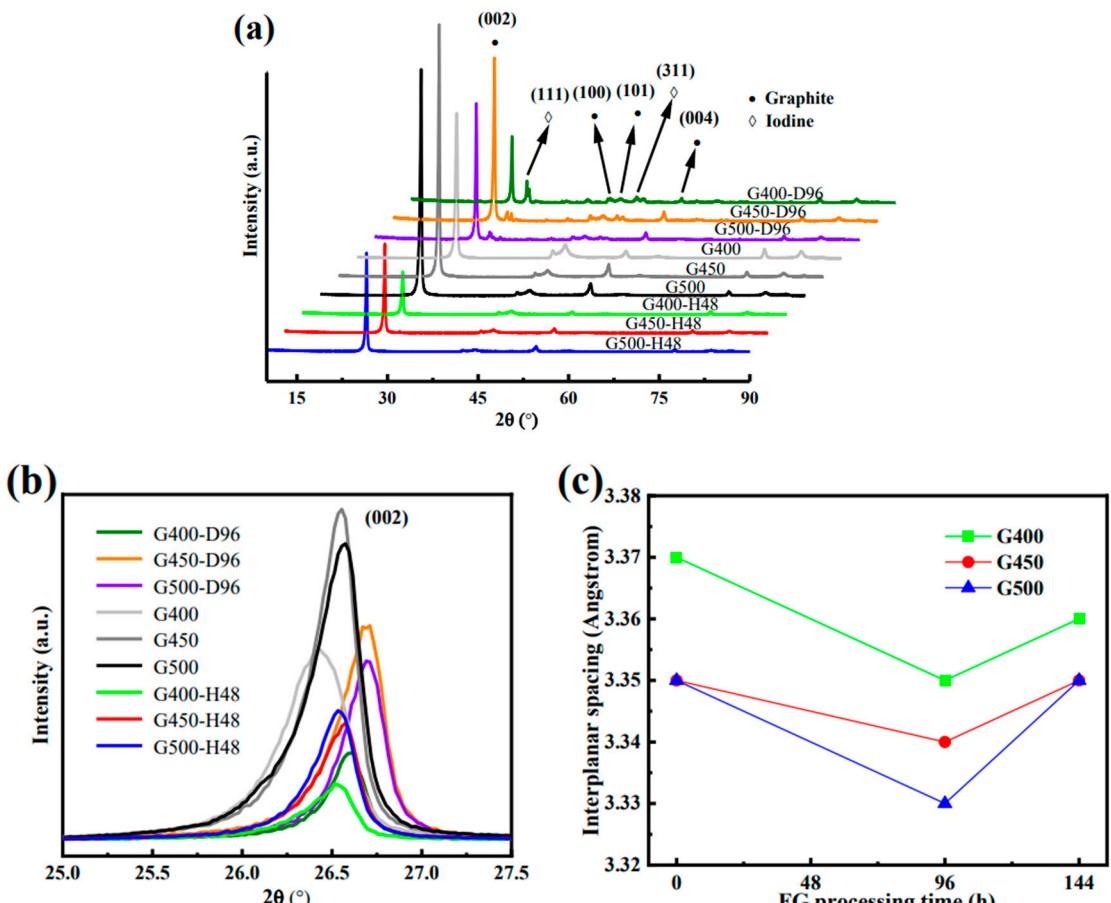

**Figure 8.** (**a**) XRD patterns of different experimental conditions, (**b**) The change trend of (002) peak, (**c**) The change trend of layer spacing of different experimental conditions.

Figure 8a,b shows the XRD spectra of microporous graphite, adsorption diffusion and isothermal annealing processes in the original state, and the presence of (111) peak is thought to be related to iodine diffusion. Figure 8c shows graphite layer spacing as iodine diffusion. As the diffusion of iodine into graphite, the interlayer spacing of graphite decreases, while after iodine diffuses out from graphite, the interlayer spacing of graphite exhibited a promising recovery. This recoverable process is more like a physical reversible process. The decrease in the spacing of graphite layers is considered to be the internal stress caused by the diffusion of iodine and the mutual extrusion between graphite layers. However, the mechanism of their internal stress interaction needs further study. The abscissa 144 in Figure 8c comes from the superposition of two diffusion experiments, namely the first 96 h of iodine diffusion experiment and the second 48 h of vacuum annealing experiment. The diffraction angle (2θ) for 002 plane increased due to iodine diffusion into graphite samples. As iodine diffuses into the graphite matrix, the layer spacing of nano porous graphite g400 and G450 decreases by 5.9 ‰ and 2.9 ‰ respectively, and the layer spacing of microporous graphite G500 decreases by 6.0 ‰. In the process of vacuum isothermal annealing, the iodine on the graphite surface diffuses out, and the interlayer spacing of graphite returns to the original state because it is no longer extruded. Therefore, the diffusion of iodine causes the diffraction angle of 002 plane to increase and the layer spacing to decrease. But when iodine diffuses out of graphite, $d_{002}$ seems to return to its original state during annealing. The decrease of $d_{002}$ indicates that iodine diffusion causes the contraction of graphite crystal in the c-axis direction. But this effect can be recovered and is more like a physical process [14]. The smaller the change in the spacing of graphite layers, the less affected the graphite structure is affected by iodine diffusion. The layer spacing change of nanoporous graphite is smaller than that of microporous graphite,

which can effectively prevent the diffusion of iodine. The spacing change of the nanoporous graphite G450 layer was the smallest (2.9‰) by comparing the spacing changes of the three graphite layers. Nanoporous graphite G450 has the best performance of blocking iodine diffusion, which is consistent with the results of previous analyses.

Raman spectra of micro-nanopore graphite samples before and after the diffusion experiments were analyzed at a laser wavelength of 473 nm and an effective penetration depth of about 50 nm. The typical D and G characteristic bands of graphite were detected in the Raman band of 1300 cm$^{-1}$ to 1700 cm$^{-1}$ [15]. The position of the D peak signal in the Raman spectrum is independent of the defect type, and its appearance only requires the defect to participate in the phonon dispersion process to meet the momentum conservation law [16]. Two characteristic bands of the D and G of pristine graphite (G400, G450 and G500) in Raman spectra appear at about 1361 cm$^{-1}$ and 1582 cm$^{-1}$, respectively. The D band strength $I_D$ of graphite depends on the defect type and density. The strength ratio ($I_D/I_G$) of D band and G band are used to evaluate the grain size and graphitization degree [17]. Based on the integral area ratio between the D band and the G band intensities, called $I_D$ and $I_G$, respectively. The following equation was used to calculate the microcrystal size $L_a$ [18]: $L_a(nm) = 2.4 \times 10^{-10}\lambda^4\left(\frac{I_D}{I_G}\right)^{-1}$.

As shown in Table 3, the microcrystalline size $L_a$ of micro/nano-porous graphite (G400, G450 and G500) decreases with iodine diffusion into this graphite. While $L_a$ increases when iodine diffuses out of graphite (annealing process of iodine-loaded graphite). The decrease in the $L_a$ value indicates that the graphite crystals are subjected to extrusion stresses in the a-axis, which corresponds to the diffusion of iodine leading to the contraction of the graphite lattice along the c-axis shown in Figure 8c. However, $L_a$ is larger after annealing, as the deformation storage energy that generated by extrusion of graphite crystals promotes further growth of graphite microcrystals at high temperatures. This may be one reason for the increase in $L_a$. In addition, another possible reason is due to the catalytic effect of iodine. During the diffusion process, iodine can adsorb on the substrate surface of graphite microcrystals and provide sufficient active sites, thereby promoting the adsorption and crystallization of carbon atoms, forming bond structures between carbon atoms, and promoting the growth of graphite microcrystals. The relative intensity ratio of $I_D/I_G$ is commonly used to characterize the defect density of graphite [19]. The increase in $I_D/I_G$ of micro/nano-porous graphite (G400, G450 and G500) after iodine diffusion is caused by the increase in defect density in graphite due to iodine diffusion. After annealing, the iodine escapes from the graphite matrix and the $I_D/I_G$ of graphite decreases, indicating a decrease in defect density. Figure 9 shows that after 96 h of iodine diffusion at 650 °C, the position of the D band shifts to the left. The iodine diffused into the pores causes internal stress between the graphite flakes of graphite, which elastically deforms the graphite microcrystals. The iodine diffuses out of the graphite matrix, and the position of the D band is recovered due to the extrusion force of graphite microcrystals is weakened after annealing. These data results indicate that iodine diffusion affects the elastic or inelastic scattering of phonons, causing the movement of the D band. The stocky Raman band between 500 cm$^{-1}$ and 1300 cm$^{-1}$ is considered a Raman characteristic band of iodine vapor because this stocky peak disappears after annealing [20,21]. This research shown that the diffusion process of iodine only causes the deformation of graphite crystals without destroying the structure of graphite.

**Table 3.** Raman shifts of D and G peaks of graphite and the ratio of the intensity $I_D/I_G$.

| Sample Name | D peak Wave Number (cm$^{-1}$) | G Peak Wave Number (cm$^{-1}$) | $I_D/I_G$ (Arbitrary Units) | $L_a$ (nm) |
|---|---|---|---|---|
| G400 | 1361 | 1583 | 0.34 | 35 |
| G400-D96 | 1353 | 1582 | 0.39 | 31 |
| G400-H48 | 1362 | 1582 | 0.28 | 43 |
| G450 | 1362 | 1583 | 0.32 | 38 |
| G450-D96 | 1356 | 1583 | 0.63 | 19 |
| G450-H48 | 1362 | 1582 | 0.23 | 52 |
| G500 | 1361 | 1581 | 0.32 | 38 |
| G500-D96 | 1354 | 1583 | 0.38 | 32 |
| G500-H48 | 1365 | 1583 | 0.20 | 60 |

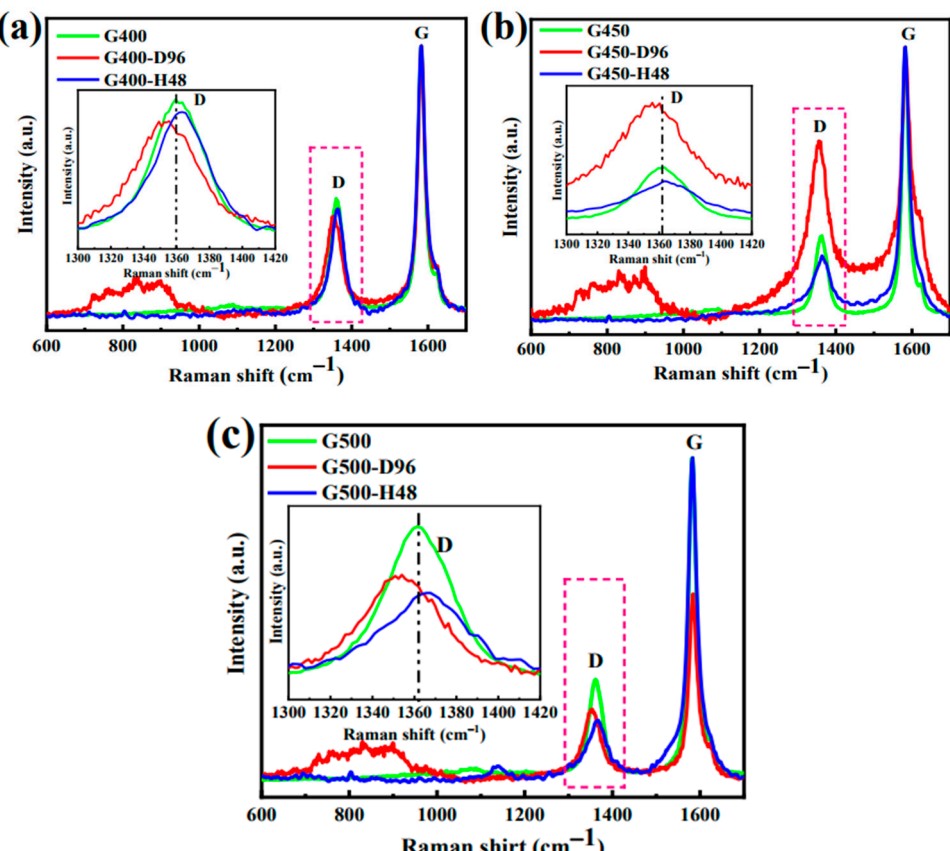

**Figure 9.** (**a**) Raman spectra of pristine, diffused and annealed nanoporous graphite G400 in the spectral range of 600–1700 cm$^{-1}$. (**b**) Raman spectra of pristine, diffused and annealed nanoporous graphite G450 in the spectral range of 600–1700 cm$^{-1}$. (**c**) Raman spectra of pristine, diffused and annealed microporous graphite G500 in the spectral range of 600–1700 cm$^{-1}$.

Figure 10 shows the intensities and wave numbers (cm$^{-1}$) of D, G and D′ bands related to all graphite samples in the Ramanspectrum. The D band is caused by the elastic scattering caused by the defect at the crystal boundary inside the graphite [22], and it is at ≈1361 cm$^{-1}$. The D′ band is relevant to the elastic scattering of vacancy defects near graphene [23], and it is at ≈1622 cm$^{-1}$. From Figure 10a,b (or Figure 10d,e or Figure 10g,h) it is known that the intensity ($I_{D'}$) of the D′ band increases first as iodine diffuses into the three kinds of graphite. The intensity of the D′ band decreases as the iodine diffuses out from the graphite interior during the annealing process (Figure 10c,f,i). Therefore, the intensity ($I_{D'}$) of D′ band increased first and then decreased with the diffusion of iodine

into and out of the graphite. Besides, the diffusion of iodine into the graphite led to the increase of the defect density. The Raman curves show the v band (Figure 10b,d) after the iodine diffuses into the graphite. As the Ferrari's stated, the presence of the v band is due to trans-polyacetylene in the grain boundary [24,25]. The phenomenon of the v band after iodine diffusion due to the diffusion of iodine into the graphite grain boundary, which affects the trans-polyacetylene in the grain boundary. This provides further evidence that iodine may diffuse into the grain boundaries of graphite, but the mechanism of the interaction between iodine and graphite is currently unknown.

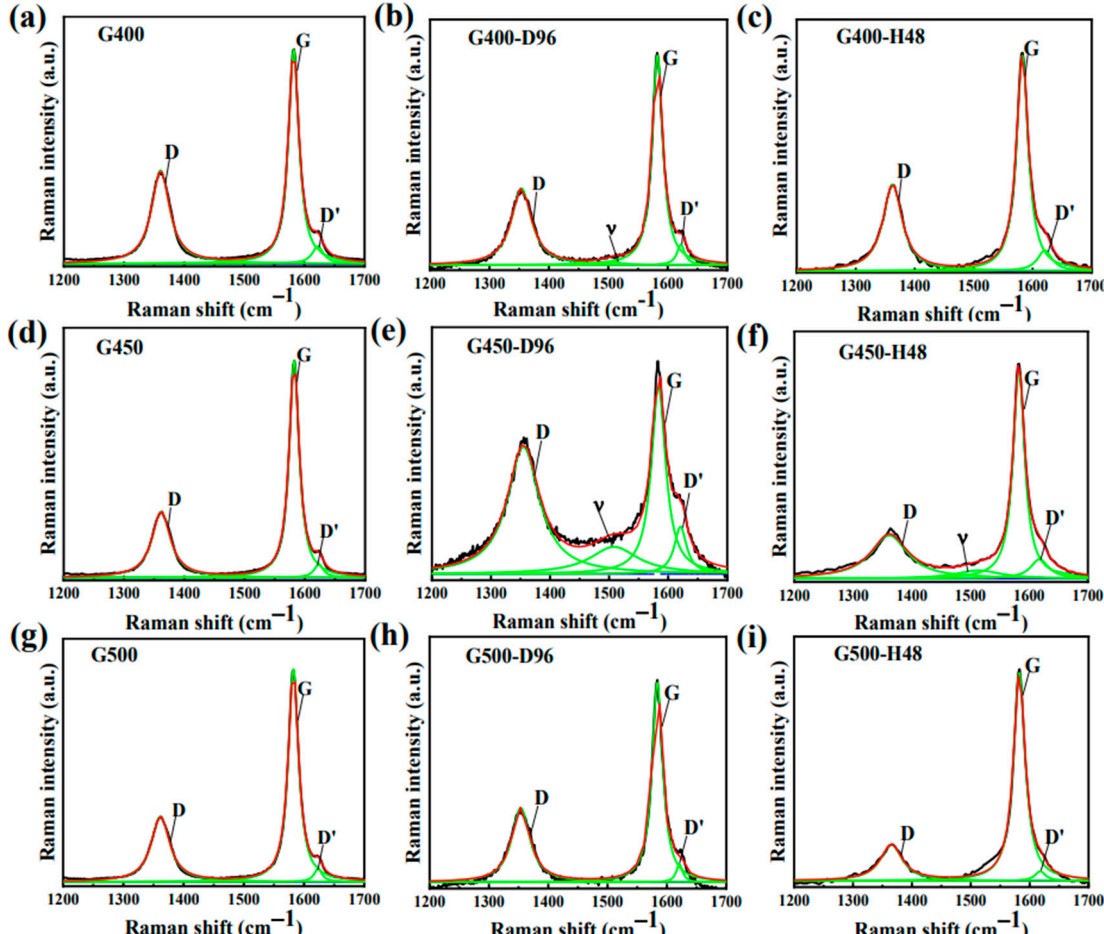

**Figure 10.** Raman spectra with linear background subtraction of (**a,d,g**) pristine micro/nano-porous graphite (Containing the G400, G450 and G500), (**b,e,h**) iodine was diffused in graphite at 650 °C for 96 h and (**c,f,i**) iodine loaded graphite is held at 650 °C for 48 h, all spectra were fitted with Lorentz lineshape fitting.

*3.4. Mechanistic Analysis*

Figure 11 shows a schematic diagram of the diffusion behavior of iodine in micro/nano-porous graphite at high temperature. This study finding that diffusion behavior of iodine in micro/nano-porous graphite is divided into three stages as followings. Firstly, iodine is captured by pores and microcracks at 650 °C and diffuses into the graphite interior along the grain boundaries. Secondly, graphite microcrystals are compressed due to iodine diffusion into grain boundaries, causing the graphite lattice to contract along the a-axis and c-axis, leading to contractions at the crystal level and decrease in lattice parameters. Finally, iodine diffuses outward along the grain boundaries or pores when the graphite sample is isothermally annealed under vacuum conditions for 48 h. The iodine content at the graphite grain boundaries decreases and the relaxation of stress with the migration of vacancy clusters. However, the graphite microcrystalline size ($L_a$) increases as the

stress decreases, and the number of defects inside graphite decreases as well. The iodine diffused from the inside of the graphite to the surface is released into the environment at the annealing process.

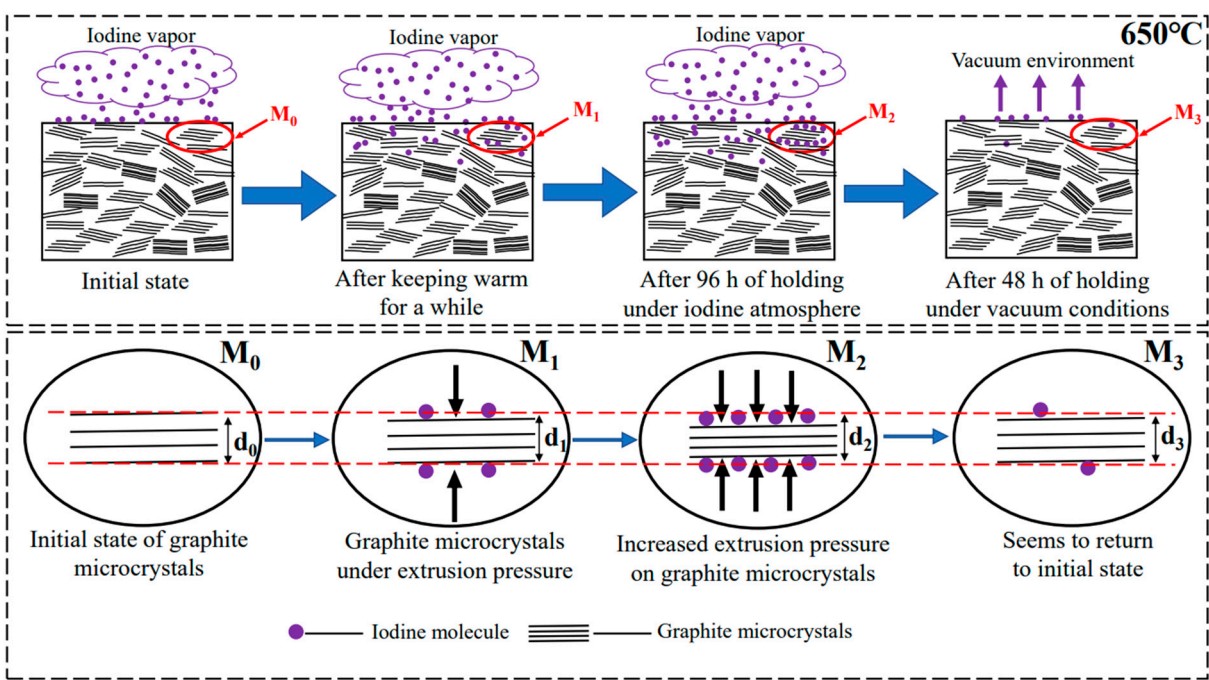

**Figure 11.** Schematic diagram of the diffusion behavior of iodine at high temperatures. $d_0$, $d_1$, $d_2$, and $d_3$ represent the changes in interlayer spacing of the original state and diffusion process, respectively.

### 4. Conclusions

At 650 °C, there is less iodine inside G400 and G450 than G500 on the condition of the diffusion depth of iodine is greater than 400 nm, and the lower iodine content in nanoporous graphite indicating that nanoporous graphite is a better barrier to iodine than microporous graphite. And due to the smaller pores, few defects and dense structure, G450 shows the best barrier performance to iodine at high temperature. The Raman spectroscopy and XRD results show that the diffusion of iodine causes the graphite crystal lattice shrinkage in the c-axis and a-axis, and increase in interplanar spacing and a higher defect density; however, when iodine diffuses out of graphite, the layer spacing of graphite seems to return to be restored to its initial state, and the defect density of graphite decreases. In addition, the graphite microcrystalline size ($L_a$) increases after annealing, which may be due to the deformation storage energy generated by the graphite crystal lattice shrinkage.

**Author Contributions:** Conceptualization, Z.-J.L. and P.-D.L.; methodology, H.-Y.Z. ang J.-L.S.; software, J.-X.C.; validation, P.-D.L.; formal analysis, Q.-B.W., J.-L.S. and Z.-F.T.; investigation, P.-F.L.; resources, Z.-F.T.; data curation, P.-F.L.; writing—original draft preparation, M.-B.Q.; writing—review and editing, M.-B.Q. and P.-F.L.; visualization, J.-L.S.; supervision, T.J.P.; project administration, Z.-J.L.; funding acquisition, T.J.P. and J.-L.S. All authors have read and agreed to the published version of the manuscript.

**Funding:** This research was funded by the National Natural Science Foundation of China (No. 52072397), ICC CAS, No.SCJC-XCL-2022-09, DNL Cooperation Fund, CAS (DNL202012), the Top-notch Academic Programs Project of Jiangsu Higher Education Institutions (TAPP) and the partial research funding was from the Priority Academic Program Development of Jiangsu Higher Education Institutions (PAPD).

**Data Availability Statement:** The raw/processed data required to reproduce these findings cannot be shared at this time as the data also forms part of an ongoing study.

**Acknowledgments:** This work was financially supported by the National Natural Science Foundation of China (No. 52072397), ICC CAS, No.SCJC-XCL-2022-09, DNL Cooperation Fund, CAS (DNL202012), the Top-notch Academic Programs Project of Jiangsu Higher Education Institutions (TAPP) and the partial research funding was from the Priority Academic Program Development of Jiangsu Higher Education Institutions (PAPD).

**Conflicts of Interest:** The authors declare no conflict of interest.

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
