# Peer review of "Diffusion Behavior of Iodine in the Micro/Nano-Porous Graphite for Nuclear Reactor at High Temperature"

_carbon, 2023_

Round 1

Reviewer 1 Report

The manuscript describes the study of diffusion of iodine in 3 different grades of graphite at high temperature using SEM, XRD and Raman spectroscopy, drawing conclusion on the effect of average pore size on the diffusion properties of iodine and the effect of iodine on graphite crystal size. 

The investigation of the migration of fission products in the structure of nuclear graphite has been presented in other studies and it is important to inform design and operational choices in the materials used in HTGR and VHTR, where nuclear-grade graphite is a moderator and structural material. Therefore, the content of this manuscript would be of interest to the readership of C – Journal of Carbon Reseach. 

While some of the results are interesting, the opinion of this reviewer is that this manuscript should undergo several major changes before being published. 

First of all, I suggest that the authors should carefully proof-read their manuscript, as there are several shortcomings in the language of the manuscript. While these shortcomings, for the vast majority, are only a minor distraction, occasionally they can make the manuscript difficult to understand and the issue should be addressed before the manuscript is considered for publication. These include spelling errors and repetitions.  

In terms of the scientific content of this manuscript, this feels very much like a “tale of 2 halves” While the second half of the manuscript, describing results from XRD and Raman spectroscopy seems sound and the conclusion drawn by the authors are based on the experimental results presented, the first half of the manuscript seems to be lacking in both terms of quality and of scientific soundness The authors studied diffusion of iodine in 3 different grades of fine grain graphite, namely G400, G450 and G500, the preparation of which is described in a previous publication by the same authors, but they do not specify which publication. My main issue with the first half of the manuscript is that the authors draw conclusions starting from what is qualitative information, mostly based on SEM observations. Diffusion involves the porous structure of graphite and the authors provide only information about open porosity and average pore size. A more detailed characterization of the surface properties and of the nano/micro porous structure of the materials is needed in order to fully understand parameters that strongly affect the effective diffusion coefficients of species through porous networks, such as surface area, surface roughness, pore size distribution and connectivity. Without knowledge of these parameters, this entire section of the manuscript feels speculative at best; interestingly, some of the conclusions drawn in this section seem to contradict some of the experimental results presented in the second half of the paper. 

A few specific comments can be found below: 

  • - I think that the naming conventions described in table 2 could have been summarized with one line of text and do not require the use of an entire table. 

  • - On line 174-175 “the diffusion of gaseous substances generally includes two processes of adsorption and diffusion”. Has the energy of the interaction between graphite’s matrix carbon and iodine atoms been studied and quantified? Can the authors comment on how this would be affected by the defect in the graphite matrix caused by irradiation in a working nuclear reactor? 

  • - Figures 5 and 6: the use of Counts of the vertical axes in all 4 figures with no indication of the magnitude of this quantity diminishes the value of the data presented in the figures. For example, having an indication of the magnitude of Counts in both figures 5 and 6 would help clarifying the next point I raise. 

  • - Lines 230-232: “At depths greater than 1.4 μm, nanoporous graphite G400 and G450 have less io-230 dine than microporous graphite G500, indicating that pore size is a key factor affecting 231 iodine diffusion performance”. While I agree in principle with the statement, if I was to base my conclusions only on the information provided, I would have expected G450 (with a smaller average pore size and overall porosity comparable to that of G400) to show less iodine within the porous structure at depth, especially since the previous figure shows that G400 and G450 contain similar amounts at depths > than 600 nm at the end of the previous experimental phase. This suggests that the mechanism is more complex and involves other parameters that are unknown. See my comment above about the vertical axis of figures 5 and 6. 

  • - On lines 309-310 the authors state: “La is larger after annealing simply because the graphite microcrystals grow further after iodine diffusion out of the graphite”. Could the authors explain why this is occurring, as it is not clear to me.

  •  

I suggest that the authors should carefully proof-read their manuscript, as there are several shortcomings in the language of the manuscript. While these shortcomings, for the vast majority, are only a minor distraction, occasionally they can make the manuscript difficult to understand and the issue should be addressed before the manuscript is considered for publication. These include spelling errors and repetitions.

Reviewer 2 Report

This paper reports the diffusion behavior of iodine in the micro/nano-porous graphite for nuclear reactor at high temperature. It is a rarely reported topic. It can be documented in C for future researchers’ references. Some questions and comments are listed below.

1.      The title describes “diffusion behavior.” But it seems it is evaporated on the surface. This title seems to be misleading. Would the authors please clarify this point?

2.      Figure 2. The experimental conditions should be briefly mentioned in the caption. This would help readers to understand the paper. For examples, (a)... (b)... (c)...

3.      Figure 3. The experimental conditions should be briefly mentioned in the caption. This would help readers to understand the paper. For examples, (a)... (b)... (c)...

4.      Figure 4. In the text, it mentioned blue and red represent carbon and iodine, respectively. But in Figure (b) (e) (h), the labels show I and C together. What does this mean? Would the authors please clarify this point?

Reviewer 3 Report

The manuscript provides valuable insights into the diffusion mechanisms of iodine in micro/nano-porous graphite and its implications for the design and optimization of nuclear graphite materials. The experimental methods employed, such as RBS, SEM, XRD, and Raman spectroscopy, are appropriate for studying the diffusion behavior of iodine. I would suggest it be published if the authors can address the following questions:

1. The study focuses on the diffusion behavior of iodine in micro/nano-porous graphite at high temperatures. How does the diffusion behavior of iodine in graphite compare to other fission products? Are there any notable differences in diffusion mechanisms or kinetics?

2. The results indicate that nanoporous graphite (G400 and G450) exhibits better inhibition of iodine diffusion compared to microporous graphite (G500). Can you provide insights into the factors that contribute to this difference in diffusion behavior? Is it primarily influenced by differences in porosity, pore size distribution, or other material properties?

Round 2

Reviewer 1 Report

The authors have addressed the majority of the issues I raised in my first review of this manuscript. The reviewer would like to thank the authors for the interesting comments and discussion in their response to my review.